META-RESEARCH ARTICLE

# Impact of gender on the formation and outcome of formal mentoring relationships in the life sciences

**Leah P. Schwartz, Jean F. Liénard, Stephen V. David** *

Oregon Hearing Research Center, Oregon Health and Science University, Portland, Oregon, United States of America

* davids@ohsu.edu

**Data Availability Statement:** All data used in this study are available from the Zenodo repository (DOI: 10.5281/zenodo.4722020).

## Abstract

Despite increasing representation in graduate training programs, a disproportionate number of women leave academic research without obtaining an independent position that enables them to train the next generation of academic researchers. To understand factors underlying this trend, we analyzed formal PhD and postdoctoral mentoring relationships in the life sciences during the years 2000 to 2020. Student and mentor gender are both associated with differences in rates of student's continuation to positions that allow formal academic mentorship. Although trainees of women mentors are less likely to take on positions as academic mentors than trainees of men mentors, this effect is reduced substantially after controlling for several measurements of mentor status. Thus, the effect of mentor gender can be explained at least partially by gender disparities in social and financial resources available to mentors. Because trainees and mentors tend to be of the same gender, this association between mentor gender and academic continuation disproportionately impacts women trainees. On average, gender homophily in graduate training is unrelated to mentor status. A notable exception to this trend is the special case of scientists having been granted an outstanding distinction, evidenced by membership in the National Academy of Sciences, being a grantee of the Howard Hughes Medical Institute, or having been awarded the Nobel Prize. This group of mentors trains men graduate students at higher rates than their most successful colleagues. These results suggest that, in addition to other factors that limit career choices for women trainees, gender inequities in mentors' access to resources and prestige contribute to women's attrition from independent research positions.

## Introduction

In academia, mentorship plays a key role as a determinant of success for both trainee and mentor [1–4]. Academic trainees spend several years training with just 1 or 2 mentors, first to obtain a PhD and then often as a postdoctoral fellow. In successful mentoring relationships, trainees develop both their intellectual expertise (through the learning of new skills and concepts) and their professional network (through the mentor's sharing of academic connections

**Funding:** This work was supported by National Science Foundation Award 1933675 (S.V.D.), https://www.nsf.gov/awardsearch/showAward? AWD_ID=1933675. The funders had no role in study design, data collection and analysis, decision to publish, or preparation of the manuscript.

**Competing interests:** The authors have declared that no competing interests exist.

**Abbreviations:** AFT, Academic Family Tree; HHMI, Howard Hughes Medical Institute; NAS, National Academy of Sciences; NIH, National Institutes of Health; NSF, National Science Foundation; PC, principal component; PCA, principal component analysis; STEM, science, technology, engineering, and mathematics.

and sponsorship). Conversely, mentors benefit in the long run from their trainees' success, as it enables a further extension of their professional networks and increases peer recognition.

At present, there is no consensus on whether the gender of the mentor influences the outcome of academic mentoring relationships. Several studies have examined correlations between mentor gender and trainee publication output, time to obtain a degree or tenure-track position, or continued interest in pursuing a career in the field. However, the literature is mixed on this point, with various studies reporting positive effects of same-gender mentoring at the graduate or undergraduate level [5–10], positive effects of mixed-gender mentoring [11], or no effect of gender on mentoring outcomes [12,13]. Proposed mechanisms linking mentor gender to trainee outcomes include increased confidence and sense of belonging in historically male-dominated disciplines among women mentored by women [9] and differences in the level of support or degree of gender bias encountered by women in research groups predominantly composed of men or women [8].

Differences in training by men versus women mentors may also reflect structural bias related to the gender of the mentor [11]. Despite some recent gains in representation, women remain underrepresented as faculty in many research fields [14] and lag behind men mentors according to conventional metrics of success. Gender-associated differences in mentor status may in turn impact the ability of trainees to perform high-impact research and thus reduce the likelihood of continuing on to formal academic mentorship roles. Such an effect would be consistent with previous research examining cumulative advantage processes in academic careers (e.g., previous results suggesting that trainees of mentors with high trainee counts tend to become mentors themselves [3,4]).

Notwithstanding the uncertainty around the impact of mentor and trainee gender on training outcomes, most studies agree that there is a tendency in mentoring relationships toward homophily, the formation of same-gender academic mentor–trainee pairs during both PhD [5,6,8,11,15–19] and postdoctoral training [20]. Surprisingly little is known about the drivers that influence homophily. In particular, it is currently unknown if there are field-level differences. Most previous work focused on a single scientific field (e.g., economics in [11,15–17]), and the few studies that encompassed several fields did not report the variation of homophily between them [13,20]. Perhaps more importantly, long-term trends in homophily within or across scientific fields have never been investigated. If gender is a meaningful driver in the outcome of mentoring relationships, then the prevalence of homophily would be an important factor shaping these outcomes.

The potential association of homophily with other characteristics of researchers, and, in particular, their relative success, is mostly uncharted. A noteworthy exception is a recent survey in life science that reported a greater tendency for men faculty that are recipients of a prestigious award to train men students and postdocs, compared to their men colleagues [20]. This study has yet to be replicated, and it is unknown if its effects generalize to other assessments of prestige.

To address these questions, we examined a multidisciplinary database of PhD and postdoc-level training relationships [3], cross-referenced with data on publication, funding, and gender (as inferred from first names). We find that gender homophily in graduate training is ubiquitous across fields, despite differences in the proportion of women students and faculty. Focusing on data from the life sciences, where sampling of mentorship data is more exhaustive, we find that both student and mentor gender are associated with different rates of trainee advancement to an academic mentorship role (i.e., a position that allows one serve as a formal supervisor to PhD students and/or postdocs). However, the effects of mentor gender on this training outcome are substantially reduced after controlling for several measurements of the mentor's status. Gender homophily in the life sciences is generally unrelated to mentor status.

However, a notable exception to this trend is the special case of scientists having been granted an outstanding distinction, evidenced by membership in the National Academy of Sciences (NAS), being a grantee of the Howard Hughes Medical Institute (HHMI), or having been awarded the Nobel Prize. This group of mentors trains men graduate students at higher rates than their most successful colleagues. These results suggest that institutional biases that affect the careers of women mentors indirectly impact the careers of their trainees and that interventions to increase representation of women as trainees may be targeted at elite scientists.

## Results

### Multidisciplinary academic mentorship dataset

We analyzed data from Academic Family Tree (AFT; available at www.academictree.org), a crowdsourced database of academic genealogy [3,21]. The database integrates user-contributed and public data on academic training relationships and publications. A mentoring relationship was either explicitly indicated by database users or inferred from authorship and supervision of a dissertation listed in ProQuest's collection of dissertations and theses.

We inferred mentor and trainee gender solely from first names. Gender inference was performed using Genderize, an algorithm that estimates the probability that a typical user of the name identifies as a man or a woman based on social media data recording how the name is commonly used [22]. Gender probabilities were available for 93.7% of individuals in AFT. Among this group, we excluded data from 4.3% of individuals whose first names did not have high probability of association with one gender (see Methods; S1 Fig and S2 Table).

We examined training relationships with end dates between 2000 and 2020, excluding data from training areas focused on business and clinical medicine (2.9% of training relationships excluded). The resulting dataset included 109,784 mentors, 23,721 postdocs, and 365,446 students from a wide range of research areas in science, technology, engineering, and mathematics (STEM), humanities, and the social sciences (S2A Fig). For training at institutions in the United States, the gender composition of graduate students and postdocs across research areas was consistent with demographic data collected by the National Science Foundation (NSF) (S2B and S2C Fig).

### Gender homophily in graduate training

Homophily, the tendency for individuals to form relationships with those similar to themselves, occurs to varying degrees for many aspects of social life (race, class, gender, age, education, behavior, attitudes and beliefs, etc.) [23]. To quantify gender homophily in mentoring relationships, we calculated the degree to which same-gender mentoring relationships exceeded the proportion expected if trainees matched to mentors randomly. Distinguishing effects of individual preferences from constraints imposed by population structure is a perennial issue in studies of homophily [23–25]. When mentors of one gender are scarce relative to students of that gender, complete homophily is impossible: the greater the scarcity, the lower the maximum level of homophily attainable (S3A Fig). We therefore normalized the value of homophily so that 0% indicates random trainee–mentor gender pairing and 100% indicates the maximum possible value, given the gender composition of the mentor and trainee pools (S3B Fig and Eq 3).

Gender homophily occurs among all general research areas we examined (Fig 1A and 1B). In all fields and all years, homophily was positive, indicating a tendency for mentors and students of the same gender to associate (median homophily across all fields and years = 20.5%). This trend is also apparent at the level of narrower fields (Figs 1C and S4 and S1 Table, median homophily across 73 fields with any women mentors = 20.3%).

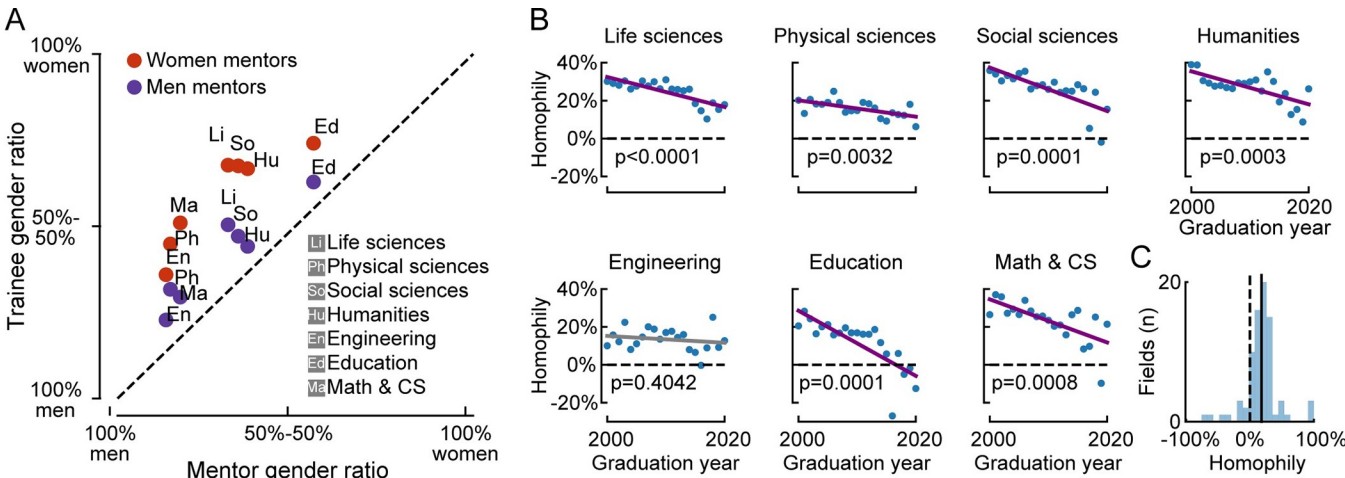

**Fig 1. Gender homophily in graduate training. (A)** Gender ratio of all PhD students from 2000 to 2020, split across broad academic fields and mentor gender. Colors indicate mentor's gender. Field abbreviations are reported in the legend. **(B)** Temporal trends in homophily, the tendency toward same-gender pairing between trainees and mentors. Each panel shows data from one major field. Lines show a linear regression of homophily as a function of graduation year. *P* values indicate significance of temporal trend. **(C)** Homophily within narrow research areas (*n* = 73 areas). Solid line indicates median. The data and code needed to generate this figure are available on Zenodo (DOI: 10.5281/zenodo.4722020).

The degree of homophily varied considerably across research areas, with the strongest homophily in humanities and social sciences and the least in physical sciences and engineering (S4A Fig). The degree of homophily within a research area was uncorrelated with its gender composition (S4B and S4C Fig). However, comparing narrower research areas with at least 1,000 students sampled showed a trend toward correlation between homophily and the fraction of women mentors or students (S5 Fig, *n* = 29 research areas, Pearson's correlation coefficient, homophily versus fraction women students, *r* = 0.36, *p* = 0.06, homophily versus fraction women mentors, *r* = 0.37, *p* = 0.05), consistent with recent work on gender homophily in coauthorship [25].

Gender homophily is decreasing over time in some fields. In 6 of the 7 broad research areas, there was a significant linear decrease in homophily between 2000 and 2020 (Fig 1B, *p*<0.05, *t* test on linear regression with time as independent variable and homophily as dependent variable). At the level of narrow research areas with more than 1,000 students, 10/29 showed a significant decrease and 19/29 showed no significant temporal trends (S6 Fig and S1 Table).

The temporal trends observed at the level of research areas can be observed at the level of mentors grouped by academic seniority. We examined the subset of mentors with at least 2 trainees and independent career start dates after 1970 (*n* = 37,962 mentors). We quantified the fraction of women mentors, fraction of women students, and homophily as a function of both mentor's career start date (1970 to 2015) and student's graduation date (2000 to 2015). The fraction of women beginning careers as mentors increased over time from 1970 to 2015 (S7A Fig). After controlling for mentor's training end date, there was no relationship between the fraction of women mentors and trainee's training end date. This result suggests that the increase in women mentors was not driven by mentor retirements between 2000 and 2015. The fraction of women students trained by the mentors also increased (S7B Fig). The decrease in homophily during this period was related to time (i.e., graduation year), but not mentors' academic age (S7C Fig).

## Gender inequity in mentor status and trainee continuation to academic mentorship roles

Consistent with previous investigations into the attrition of women across the academic career track (sometimes known as the "leaky pipeline") [14,26–28], our results show that the proportion of women in social science and STEM fields is lower at progressively later stages of the academic career track, from graduate student to postdoc to academic mentor (S11A Fig). This result indicates the population of academic mentors remains skewed toward men, even in research areas with student populations close to gender parity. However, it does not in itself indicate whether women graduate students continue on to academic mentorship positions at the same rate as men graduate students. In addition, it does not indicate whether structural gender biases that affect women as mentors indirectly affect retention of their students in academia as formal mentors.

To address these questions, we examined the proportion of graduate students and postdocs in the life sciences that continued on to academic mentorship, accounting for factors that may impact continuation (see S1 Appendix). We hypothesized that if men and women mentors differ in status (defined here as access to funding, labor, and prestige markers such as citations) due to gender bias, these disparities might lead to differences in trainees' continuation to academic mentorship roles. We therefore compiled several widely used metrics to quantify mentor's status: $h$-index (a measurement of citation rate and publication production [29] based on data from the National Library of Medicine and Semantic Scholar), trainee count (total number of PhD students and postdocs mentored, a metric closely related to laboratory size [3]), the rate of funding granted by the US governmental agencies NSF and National Institutes of Health (NIH), and the rank of the mentor's academic institution in the Quacquarelli Symonds World University Rankings, an annual assessment that heavily weights the institution's reputation among academics. Funding rate, $h$-index, trainee count, and institutional prestige were all correlated with one another, suggesting that all 4 metrics measured a common trait of "aggregate status" (S8 Fig). To compare mentors of the same status, we sorted mentors of all genders by each status metric, then grouped them into up to 10 bins of approximately equal size, such that mentors with the same value for a status metric were never placed in different bins.

We limited our analysis of mentor status to the life sciences. Sampling was more complete in these fields for 2 reasons. First, data on publication and funding were drawn from sources specific to this field (including funding data from the NIH and publication data from the National Library of Medicine). Second, because the AFT began as an effort to crowdsource the academic genealogy of neuroscience, its sampling of mentorship data is most dense for the life sciences. To avoid false negatives in our identification of trainees that continued to academic mentorship roles, we further limited the analysis to the subset of training relationships with stop dates before 2010 and whose records had been manually edited by AFT users (final $n = 11,112$ mentors, 26,420 trainees, 26% of life science training relationships meeting other criteria for analysis). Although these criteria reduced the size of the dataset, they minimized the chance of false negatives in our identification of progression to academic mentorship. Due to our strict definition of continuation as progress to academic mentorship, it is likely that the continuation rates reported here (see S1 Appendix) underestimate the actual proportion of trainees that remain in academia.

Compared to women mentors, men mentors had higher mean rates of funding, trainee count, and $h$-index, but not institution rank (Fig 2A, $p < 0.00$, Welch's unequal variances $t$ test). Consistent with this finding, men mentors were overrepresented at the highest status deciles for funding, trainee count, and $h$-index, while women mentors were overrepresented in lower status deciles (Fig 2B).

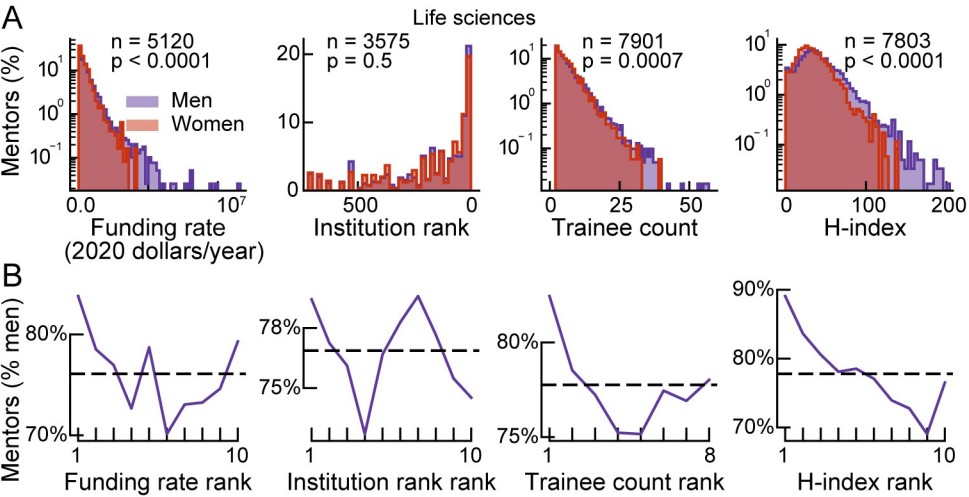

**Fig 2. Relationship between mentor gender and mentor status.** **(A)** Distribution of mentor-status metrics among men and women mentors in life sciences (*n*: total number of mentors in the continuation dataset with data for the status metric, *p*: *p*-value of Welch's unequal variances *t* test for difference in mean between men and women mentors). For improved visualization, 2 outliers are not included in the histograms (trainee count = 75, *h*-index = 230). **(B)** Gender distribution of mentors after binning by status (smaller numbers higher rank). Solid line indicates percentage of men mentors within each bin. Dashed line indicates percentage of men mentors across all bins. The data and code needed to generate this figure are available on Zenodo (DOI: 10.5281/zenodo.4722020).

To test the hypothesis that structural gender bias among mentors indirectly affects the rate at which trainees continue to academic mentorship positions, we fit logistic regression models that predicted student and postdoc continuation based on each mentor status metric individually, trainee and mentor gender, mentor seniority, and training end date (Fig 3, left). Including

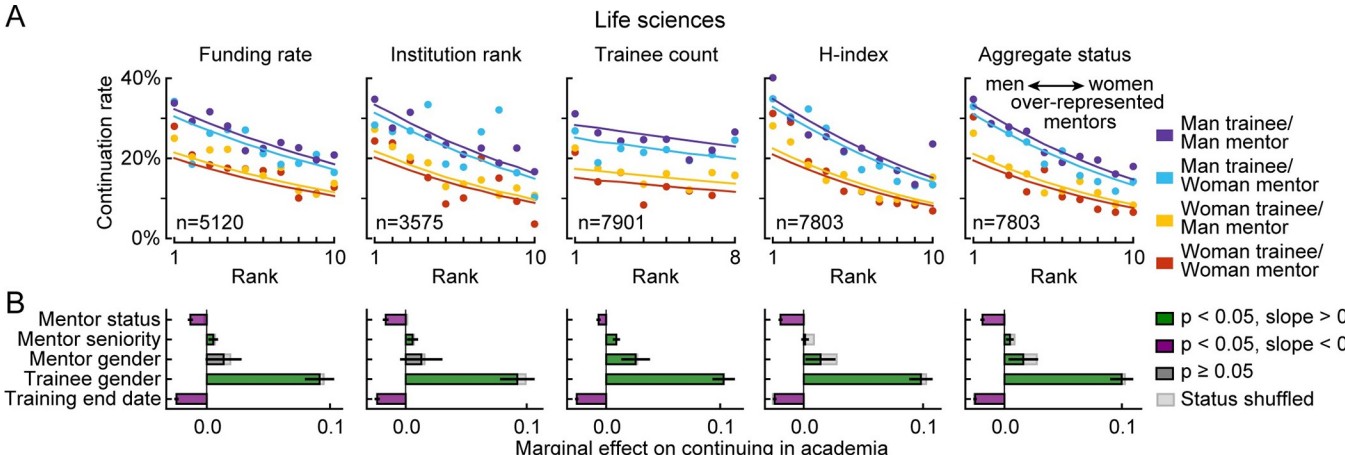

**Fig 3. Association between mentor status, mentor gender, and trainee continuation in academia.** **(A)** Mean continuation rate in academia for PhD students and postdocs in life sciences. Each panel shows data sorted according to a different measure of mentor status. Points show mean continuation rate of trainees of mentors with a given status (smaller numbers indicate higher status), grouped by trainee and mentor gender. Lines show prediction of logistic regression model, fit to these variables as well as training end date and mentor seniority. Titles indicate number of mentors with respective status data available. **(B)** Marginal effects of each independent variable on trainee continuation rate, predicted by logistic regression models incorporating the status variables in (A). Marginal effects of gender show the impact of the mentor or trainee being a man relative to a woman. Marginal effect of mentor status shows impact of a 1-decile increase (worsening) in rank compared to others in the field. Marginal effect of mentor seniority shows impact of a 10% increase in the variable. Error bars show 95% confidence intervals. Light gray bars show marginal effects for same model fit to data with mentor status shuffled across trainees. The data and code needed to generate this figure are available on Zenodo (DOI: 10.5281/zenodo.4722020).

training end date as an independent variable accounted for long-term changes in the number of trainees, status variables and continuation rates [30]. Mentor seniority was included to control for the possibility that phenomena apparently related to gender disparities in mentor status could be explained instead by gender differences in mentors' academic age. We also fit a model that included the first principal component (PC) of all 4 status metrics as a single "aggregate status" variable (Fig 3, right). To quantify the degree to which differences in mentors' status account for differences in trainee continuation rates, we compared each model to one in which mentor status had been shuffled across trainees (Figs 3 and 4).

For all measures of mentor status, higher rank was associated with greater rates of trainee continuation to roles as academic mentors (note consistently sloping lines in Fig 3). Being a man or the trainee of a man was also associated with greater continuation rates. However, this disparity was substantially reduced if one considered the overrepresentation of men in higher mentor status ranks (Fig 2). Including a measure of mentor status in the model substantially reduced the effect of mentor gender. For all measures of mentor status, the magnitude of the mentor-gender effect was reduced relative to a model in which mentor status was randomized. This randomization had minimal impact on trainee-gender and temporal effects (Figs 3B and 4). Thus, controlling for mentor status reduces the apparent effect of mentor gender on trainee retention as academic mentors by up to 49% ($p < 0.002$, $t$ test, for all metrics except trainee count, Fig 4C). The maximum reduction occurred in the model that included $h$-index, and the aggregate status metric did not have greater predictive power than $h$-index.

Incorporating status metrics also accounted for some effects of mentor seniority (Fig 4B). In a stepwise comparison, randomizing data for both status and seniority resulted in a greater reduction of mentor-gender effects than randomizing either alone, suggesting that gender differences in seniority do not account for effects of gender differences in status (S9 Fig).

Separately analyzing data for graduate students and postdocs (S10 Fig) showed consistent effects for mentor status, mentor seniority, and trainee gender. Mentor-gender effects did not reach significance among all subsets of the data, possibly because of the reduced statistical power available in these smaller datasets.

## Gender homophily and mentor status

The differences in status between mentors of different gender (Fig 2) suggests that gender homophily will pair trainees with mentors whose structural advantages or disadvantages reinforce their own [31]. This effect may be reduced or amplified if homophily differs across mentor status levels. We therefore analyzed how homophily and trainee gender differ according to mentor status. In addition, we separately analyzed data for highly elite mentors, as indicated by a Nobel Prize, membership in the NAS, and/or funding by the HHMI. It is well established that research communities are biased to value the output of researchers who have established a high profile [32]. Moreover, it has been reported that men mentors in this group train fewer women than their peers, possibly exaggerating the impact of homophily in this group [20].

Higher status—as measured by funding rate, $h$-index, trainee count, institution rank, or the aggregate status metric—did not show a consistent relationship with the percentage of men students trained by men or women mentors (Fig 5A). The degree of homophily, which collapsed across both men and women mentors, did not show a relationship with mentor status for any individual status metric or the aggregate status metric (Fig 5B, $p \geq 0.05$, linear regression predicting homophily from mentor status bin).

By contrast, mentors that received a prestigious award tended to train more men students than high-status mentors that were not prizewinners (Fig 5A). The percentage of men trained by mentors that received prestigious awards was greater than mentors at the highest status

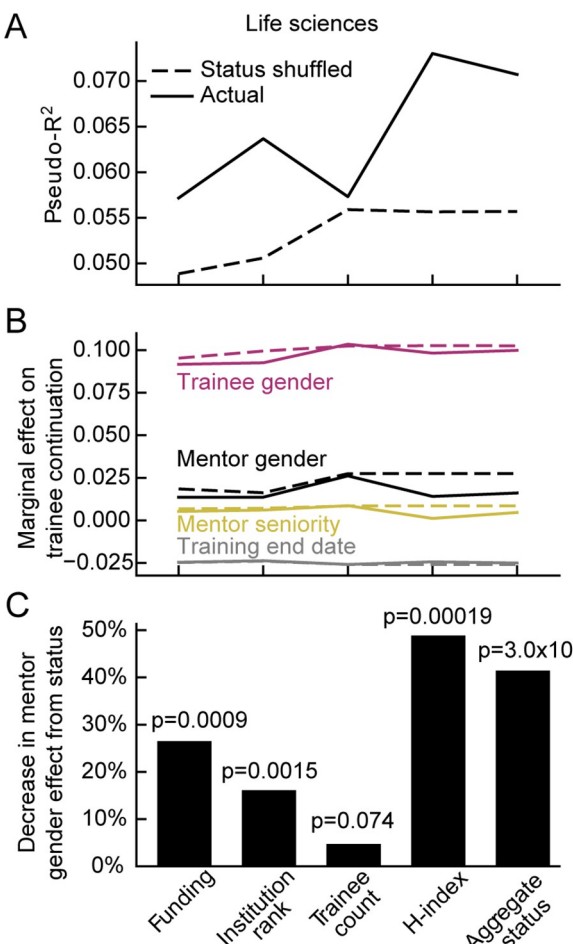

**Fig 4. Reduction in mentor-gender effects after controlling for mentor status. (A)** Performance of logistic regression model predicting trainee continuation based on training end date, mentor seniority, trainee and mentor gender, and mentor status before (solid line) and after shuffling (dashed) mentor status across trainees. Models are fit to the subset of life sciences data with information available for the corresponding status metric (see Fig 3). Performance of the status shuffled models varied because the pool of mentors with available data varied between metrics. **(B)** Marginal effect of gender and temporal variables for models in which mentor status is shuffled (dashed line) or is not (solid lines), quantified as in Fig 3B. **(C)** Percent decrease in marginal effect of mentor gender after incorporating each mentor status variable into the logistic regression model ($p$ indicated for $t$ test between marginal effect for models with and without each status metric shuffled. The data and code needed to generate this figure are available on Zenodo (DOI: 10.5281/zenodo.4722020).

aggregate status rank (50% men in top decile versus 61% among awardees, $n = 2,490$ mentors, $p = 6.1 \times 10^{-22}$, $t$ test). This trend was observed among both men and women mentors (men mentors: 53% men in top decile versus 65% among awardees, $n = 1,991$ mentors, $p = 4.3 \times 10^{-21}$, $t$ test, women mentors: 36% versus 47%, $n = 499$, $p = 1.7 \times 10^{-6}$). The fraction of men trained differed between award recipients and mentors ranked in the top decile according to several status metrics, suggesting that this finding generalizes across multiple measurements of prestige (funding rate: 49% men in top decile versus 61% among awardees, $n = 2,075$ mentors, $p = 8.0 \times 10^{-21}$, $t$ test, trainee count: 44% versus 61%, $n = 2,480$, $p = 3.0 \times 10^{-53}$, $h$-index: 51% versus 61%, $n = 2,480$, $p = 2.3 \times 10^{-15}$, institution rank: 51% versus 61%, $n = 1,844$, $p = 1.3 \times 10^{-11}$).

To more rigorously test the interaction between award receipt and the gender composition of student trainees, we turned to multivariate regression. A regression predicting fraction of

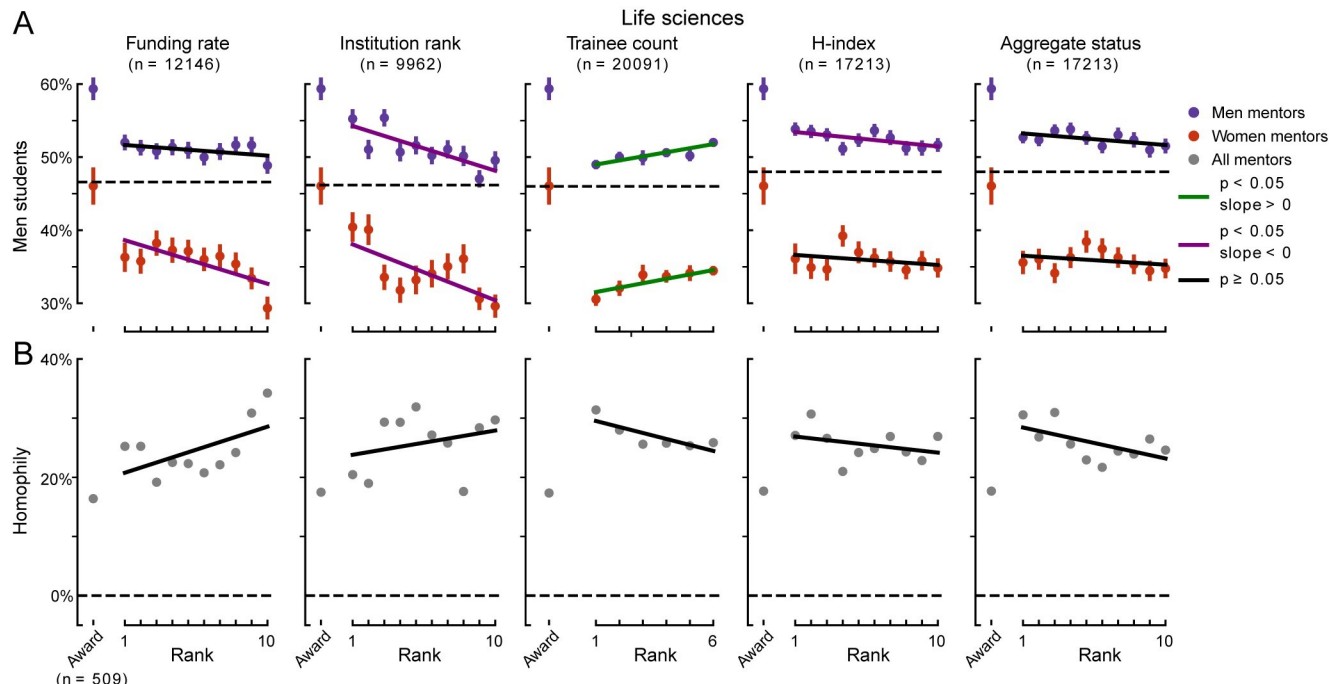

**Fig 5. Relationship between mentor status and gender homophily.** (A) Mean percentage of men students among life sciences mentors divided by research area, mentor status, and acquisition of elite awards. Horizontal dashed lines indicate percentage of men students for all mentors. Color of solid lines indicates significance of a linear regression predicting the percentage of men students for each mentor, based on decile rank in status and receipt of an award (see legend). Error bars show SEM. Numbers ($n$) indicate total mentors with valid status measures for that group. (B) Mean homophily among life sciences mentors, grouped as in (A). Color of solid line indicates result of linear regression predicting homophily based on mentor status. The data and code needed to generate this figure are available on Zenodo (DOI: 10.5281/zenodo.4722020).

men students for each life science mentor based on award receipt, mentor gender, and success rank showed an effect for award ($\beta = 0.077$, $p = 2.1 \times 10^{-7}$) and mentor gender ($\beta = 0.16$, $p < 1 \times 10^{-29}$, $n = 17{,}205$ mentors) but only a trend for status ($\beta = -0.0017$, $p = 0.057$).

## Discussion

Our results indicate that graduate-level mentoring relationships are formed at higher rates between trainees and mentors of the same gender. This phenomenon of homophily occurs consistently across wide-ranging research areas. Gender groups are also associated with differences in mentoring outcomes, as measured by trainees' continuation to positions that allow for formal supervision of doctoral students and postdocs. Most prominently, women graduate students and postdocs have lower continuation rates than men, consistent with many previous reports [14,26–28]. We also find that continuation rates in the life sciences depend on mentor gender, as trainees of men mentors have higher continuation rates than trainees of women mentors. However, we find a substantial portion of this latter association is accounted for by the observation that men mentors in life sciences also have higher average rank in traditional measures of status. These findings support a model in which mentors' access to funding and labor, as well as prestige markers such as citations, are distributed unevenly by gender and in turn affect trainees' retention in academia as formal mentors. Compounded with gender homophily, these disparities in mentor status may contribute to ongoing challenges around the retention of women trainees in academic research.

## The leaky pipeline phenomenon is widespread across fields

Our finding that rates of continuation to academic mentorship are lower for women graduate students and postdocs replicates prior research that has identified postgraduate career transitions as points of women's attrition from academia, although we note that there is a distinction between acting as formal mentor, holding a tenure-track position, and participating in academic research in any capacity [26–28,33]. Although the fraction of women graduate student trainees increased between 2000 and 2010, the magnitude of "pipeline leak" did not change. Numerous factors have been proposed to explain this phenomenon [34–38]. In addition to bias in assessment and hiring, women may experience greater obligations to family and childcare relative to men or lack of institutional support for balancing family and career. Our data do not directly address the relative role of these and other factors in causing the leaky pipeline, which remains a topic of debate. However, we do show that disparities in retention associated with trainee gender persist even after controlling for graduation year (a proxy measure of the increasing competitiveness of the academic job market [39]) as well as (for the specific case of the life sciences) several measurements of mentors' academic achievement. This result indicates that student gender is associated with differential outcomes even when comparing students that are similarly situated when training ends. In addition, our data indicate that attrition of women during postgraduate career transitions occurs across multiple area of STEM and social science and ultimately affects the size of the pool of women graduate mentors.

## The role of academic mentor gender in trainee retention

We offer evidence that gender-based disparities in the status of mentors in the life sciences contribute to disparities in the rate at which their trainees continue to academic mentorship roles. Because our sources for data on citation, funding, and mentor receipt of an award are most complete for the life sciences, our results on mentor status are limited to this field, unlike our results on homophily and the leaky pipeline phenomenon, which spanned multiple fields.

Our results are consistent with previous findings of disparities between men and women faculty on several conventional measures of success. These include lower number of publications by women researchers [6,11,14], lower rates at which their work is cited [40], and lower levels of funding [41–45]. Gender differences also extend to slower progression through professional positions [26,46–49], as well as more subtle markers of career achievement such as invitations to present work at seminars [50].

The causes of gender-based disparities in conventional measures of academic success are complex. They may reflect long-standing bias in the academic community, which leads women's competence or performance to be assessed on a different scale from men or according to different qualities [34]. Evidence of bias has been found in studies of gender differences in the outcomes of fellowship applications [41], hiring assessments [36], and the content of reference letters [51,52]. Effects of bias may be amplified by constraints and cultural expectations related to family life, particularly childcare, which impact women more than men researchers, especially in early and mid-career stages [14,27,53]. Recent longitudinal studies indicate that parenthood impacts women's participation in the STEM workforce more than men [38] and accounts for a large share of gender differences in research production [54]. Geographical constraints due to partners' professional employment may also impact women in academia to a greater degree than men [55].

Proposed mechanisms linking mentor and trainee gender to training outcomes include increased confidence and sense of belonging in a historically male-dominated field among women mentored by women [9]. Students have also reported qualitatively different support depending on gender. Specifically, women graduate students are more likely to report

benefiting from psychosocial support from a mentor during their PhD training (such as providing emotional support and taking an interest in the student's personal life, cf. [56]), whereas men are more likely to report operational support (such as being involved in chairing a conference, collaborating on papers, or being recommended to colleagues, cf. [57–59]). It is not clear from this descriptive evidence whether such differences in support are driven by mentor preferences, trainee needs, or effects of gender stereotypes on trainee expectations of mentors [60]. Women faculty generally assume a greater share of academic service responsibilities than men [61], and differences in mentoring demands may add to this disparity.

Future research could test whether covariates of mentor gender, in addition to status, affect retention in academia as mentors. For example, trainees of men and women mentors may be judged differently, even if the mentors have similar qualifications, reflecting a general gender bias in academia [34,36] (but see [37]). Men and women mentors' social networks may differ in gender composition, as suggested by analyses of coauthorship [25,62], or in other features such as size. Finally, gender differences in self-promotion styles (e.g., self-citation and use of positive language to describe research results) may be imitated by trainees [63,64]. Future research could also examine whether subfields or topics within the life sciences differ in the relationship between mentor gender, mentor status, and trainee continuation.

Several studies that have documented positive effects of same-gender mentoring for women students are compatible with our specific result. In particular, our findings are compatible with evidence that contact with women role models has positive effects on women's persistence in STEM careers at stages prior to graduate school [9,10]. Our results are also compatible with evidence that women's careers benefit from a social network that includes women during graduate education [65]. Our results differ from a recent study that found higher continuation rates among women in chemistry that work with women PhD mentors, after controlling for students' research productivity [8]. However, our general finding that apparent differences between men and women mentors are reduced after controlling for status is consistent with a recent preprint on gender and research productivity [66]. We view such results, like our own, as reason to work for equity in institutional support for women mentors in academia.

Given the many factors that can lead to gender disparities, it is unlikely that equity can be achieved with a single policy change. For example, analysis of NIH data suggests that outcomes of grant review are not biased against women applicants [67]. Still, a disparity in funding levels persists, which must reflect differences in the rate at which women apply for funding or in the size of their awards. In this study, the status variable that accounted best for the disparity in mentorship outcomes was $h$-index. While $h$-index can be criticized as overly narrow in its scope, this observation is consistent with the idea that a researcher with high publication impact is likely to be well known and respected within their field. By this logic, investment by universities and professional societies in programs that enable women mentors to increase the impact of their work in ways that increases metrics such as $h$-index may also benefit trainees seeking positions as academic mentors.

## Homophily in academic mentorship

A recent survey of life science researchers suggests both students' and mentors' preferences influence gender homophily in mentoring: Applicant pools are skewed toward the gender of the mentor, but the gender composition of the final research group matches the mentor's gender more closely than the applicant pool [19]. Comparisons of homophily across differing levels of organization, such as the subfield, department, or research group could help to expand this picture. For example, researcher gender composition varies across subfields [68]. If a subfield contains many students and mentors of one gender, this would increase the degree of

homophily within the field of which it is a part (see [25] for similar observations on homophily in coauthorship). Gender differences in subfields may be influenced by the degree to which particular research topics or methods fit with internalized gender stereotypes, or whether the culture of the subfield makes students of a particular gender feel that they belong [69,70]. Comparisons of gender homophily across subfields could therefore reveal the extent to which drivers of gender homophily lie outside the process of applying for research supervision. A study of gender homophily in coauthorship of life sciences articles found only a weak relationship between the degree of homophily and the journal's discipline [25], suggesting that homophily is driven by choice of coauthors rather than gender differences in research topic choice. Our finding that there is only a weak correlation between homophily and research area suggests similar causes for gender homophily in mentorship. Another possible influence on homophily is the research group itself, which is a source of informal mentorship, acculturation, and support, particularly in STEM. A sense of affinity for the research group could also draw students to work with particular mentors. It would therefore be interesting to know whether there is more or less gender homophily in fields where mentoring is more one-on-one. Future research could also examine how gender homophily interacts with other demographic variables, particularly race and ethnicity.

We show that homophily is unrelated to markers of mentor status, with the exception of receipt of a prestigious award. Our results thus build on the finding that elite men mentors in the life sciences employ fewer women than their colleagues [20], suggesting that it does not generalize to other indicators of prestige (as it would if we had found a consistent decrease in the percentage of men students across success levels). Instead, mentors that receive prestigious awards are a special case, but an important one, given their role as feeder labs for independent researchers [20].

## Limitations of the current study

Given that we rely on observational data, our ability to identify causes is limited. We have attempted to use appropriately qualified language to describe our findings, and to discuss their relationship to controlled experiments on gender. However, we also note that there is a rich tradition of using observational data and statistical models to study how gender affects academic careers under real-world conditions. A critical aspect of this approach is to identify underlying factors that explain differences between observed groups [71]. The analysis of mentor status illustrates this approach, where an apparent effect of mentor gender can be explained by differences in the resources and prestige associated with men versus women mentor groups. As models are refined with more detailed and quantifiable variables, they may be used to drive experiments that test causal relationships.

Static, binary gender categories are a simplification of the complex social and biological reality of sex and gender [72,73]. Due to the probabilistic nature of the methods used in this study, we were unable to identify transgender, intersex, and/or nonbinary individuals in the data. There is evidence that transgender status influences experiences in academia. An account by a prominent transgender scientist indicates that gender transition affected his treatment by colleagues [74]. Survey data indicate that transgender graduate students experience stress in day-to-day interactions with peers and faculty due to their gender identity [75,76]. We hope that future research on gender and mentoring will integrate findings from studies that leverage the large sample sizes available through automated analysis of first names with analysis of survey data that incorporates more complex understandings of gender. Ideally, those studies would consider the diversity of experiences within the transgender population (e.g., transmasculine and transfeminine, age and career stage of gender transition, nonbinary and binary).

An important limitation of this study is its focus on placement in positions that enable formal supervision of doctoral students and postdocs as a measurement of training outcome. Formal academic mentorship does play an outsized role in shaping research communities and thus merits special attention. At the same time, this mentorship role is only one of many different and valuable outcomes that academic training makes possible. Individuals that go into a teaching-focused job at a non-PhD granting institution may have an impact on undergraduates' persistence in STEM [9,10] or other fields. Trainees may also directly advance scientific knowledge through work in industry or conducting and supervising research at non-PhD granting academic institutions. We focus on doctoral and postdoctoral supervisors because of the role they play in defining research problems and training the next generation of faculty, as well as the possibility that gender disparities in this specific trajectory may be an indicator of more widespread sexism in academia.

## Materials and methods

Data for the current study were drawn from the AFT (https://www.academictree.org), an online database of mentoring relationships [3,21]. The database records the identity of the mentor and trainee, the type of training (graduate or postdoctoral), and the start and end year of the training.

The AFT derives information on training relationships from 2 sources: crowdsourced (i.e., user-provided) data and ProQuest's collection of dissertations and theses. In a previous study, a portion of the crowdsourced data was validated by comparison with data on formal mentoring relationships indicated on faculty web pages [21]. To limit heterogeneity in the type of mentoring relationships included [77], ProQuest data were limited to records of dissertations that resulted in doctoral degrees within a recent time period (2000 to 2015) and listed the dissertation advisor. When the name and institutional affiliation of an advisor listed in ProQuest matched the name and field of study of a mentor included in crowdsourced data, the two were considered to refer to the same person. If no name match was found at any institution, a new node was added for the advisor. If a name match was found at another institution, the training relationship was not added until it could be reviewed manually as being a match to an existing mentor or a new mentor.

As of November 2020, the AFT contained data on 724,657 researchers and 695,045 training relationships. Data for 397,008 training relationships (57%) were populated based on the ProQuest dissertation database. Trainees and mentors for existing AFT data were filled in based on name and institutional affiliation matches to the ProQuest data.

We analyzed data from 79 labelled research areas (S2 Fig). Because these labels for areas are added to the AFT by public contributors, the size and specificity of their respective research communities varies. We therefore grouped data into 8 broad fields based on the categories used in the NSF's Survey of Earned Doctorates (S2 Fig).

There is a correlation between the fraction of PhD-level training relationships in each broad field in the AFT dataset and concurrent data from the NSF's census of PhD-level training relationships ($r = 0.97$, $p = 0.0003$, AFT data limited to training relationships at US institutions). Thus, the sampling in the overall AFT dataset matches that reported by the NSF. Examining crowdsourced data alone shows an overrepresentation of the life sciences and physical sciences relative to their proportion in NSF data. This trend may be due to the history of AFT, which began as an effort to track the academic genealogy of neuroscience through crowdsourcing [21].

Geographic locations were available for 91.8% of training relationships and indicated that the vast majority (90.2%) of data was drawn from US institutions. The gender composition of

US graduate students across research areas was highly correlated with equivalent data from the NSF Survey of Earned Doctorates, an annual demographic study of US graduate programs (Pearson's correlation coefficient, $r = 0.97$, $p = 0.0003$, S2B Fig). Comprehensive demographic data on postdocs by gender, field of study, and year of training end date were not available. However, the gender composition of US postdocs in STEM and social sciences was correlated with data from the 2015 NSF Survey of Graduate Students and Postdoctorates in Science in Engineering, a cross-sectional survey ($r = 0.98$, $p = 0.003$, S2C Fig).

Data analysis was implemented in Python and R [78–83]. The subset of AFT data analyzed in this project, along with code for producing figures, is available on Zenodo (DOI: 10.5281/zenodo.4722020).

## Gender inference

Researcher gender was inferred from first names using genderize.io, an online portal that relies on social media data to estimate the probability that a first name is associated with an individual identifying as a man or a woman [22]. Statistical analysis of authors' first names have previously been to used to study gender differences in academic publication and citation [40,68,84,85]. In previous evaluations, it has been shown to have high levels of accuracy when applied to editorial boards of academic journals [86] and author lists [22,85]. Because some names show differences in their typical gender (e.g. "Robin" is typically a man's name in the United Kingdom, but not in the US), usage data for gender inference were drawn from the country in which an individual's academic institution is located. When no location information was available, we used gender estimates based on usage data pooled across all countries.

We excluded data for researchers whose names were not clearly associated with one gender. To avoid possible bias in measurement error [84], we adjusted the thresholds slightly for assigning gender to names to balance the probability of error for men and women. Our procedure for adjusting the threshold was as follows:

(1) For each probability $p$ (0 = woman, 1 = man), find the number of names, $n(p)$. (2) For each probability, calculate the number of names expected to be falsely assigned to each gender (e.g., for men expected error at probability $p = n(p) - p^*n)(p)$). (3) Using the results from (2), calculate the cumulative probability of error if the threshold is set at each possible value of $p$. (4) Choose a threshold to use for one gender. (5) Using the results from (3), find the corresponding cumulative probability of error for one gender at this threshold. Choose the threshold for the remaining gender that gives an equal cumulative probability of error.

Following this procedure with a fixed threshold of $p \geq 0.75$ for including putative men results in a threshold of $p \leq 0.24$ for including putative women (see S1 Fig). A comparison with stricter thresholds on key results showed no consistent difference between fixed and balanced thresholds (S2 Table).

We manually validated the gender estimated through genderize.io on a randomly chosen subset of $n = 2,184$ researchers with a profile picture on the web portal of the AFT. Specifically, each profile photo was presented to 2 different scorers (out of 3 total scorers) who were instructed to report the apparent gender of the person, in the absence of any other clue beside the picture. The scorers could report one of the 3 options: "male," "female," and "ambiguous/hard to tell from this picture". Gender probabilities based on first names were available for 2001 researchers in the sample. We excluded pictures reported as ambiguous by one or both scorers ($n = 24$ reported ambiguous by both scorers, $n = 189$ reported as ambiguous by one scorer). There were no instances in which one scorer perceived the individual in the photograph as a man and the other perceived the individual as a woman. Scorers generally marked photos as ambiguous due to technical errors in loading the photo rather than uncertainty

about the individual's gender presentation. We found a high rate of agreement between classification based on first names via genderize.io and scorers' reports based on photos ($n = 1{,}788$ researchers, area under ROC curve = 0.99). A detailed breakdown of error rates is provided in S3 Table. The magnitude of error rates is similar to those reported in a previous study that used algorithmic identification of gender based on names [84].

## Homophily

We measured homophily as the degree to which same-gender mentoring relationships exceeded the proportion expected if trainees matched to mentors randomly. Homophily was first calculated separately for men and women:

$$\text{homophily}_F = Pr(\text{trainee}_F | \text{mentor}_F) - Pr(\text{trainee}_F) \tag{1}$$

$$\text{homophily}_M = Pr(\text{trainee}_M | \text{mentor}_M) - Pr(\text{trainee}_M) \tag{2}$$

Overall homophily was then computed as their sum, weighted by the total number of training relationships with mentors in each group:

$$\text{homophily}_{total} = Pr(\text{mentor}_F) * \text{homophily}_F + Pr(\text{mentor}_M) * \text{homophily}_M \tag{3}$$

Positive values indicate that students and mentors of the same gender tend to work together, while negative values indicate that students and mentors of different gender tend to work together. A value of 0 indicates that students of any gender have an equal chance of training with mentors of any gender. Values of homophily were normalized so that 100% indicates the maximum possible value, given the gender composition of the mentor and trainee pools.

To demonstrate effects of the gender composition of a research field on homophily, we conducted a simplified simulation (S3 Fig). We assumed that all individuals in the field had a set propensity to form same-gender mentor–trainee pairs, described by a parameter ($a$) between 0 and 1. This parameter determined the initial number of man–man and woman–woman mentor–trainee dyads in the field:

$$n(M \text{ trainee}, M \text{ mentor}) = a * \min[n(M \text{ trainee}), n(M \text{ mentor})] \tag{4}$$

$$n(F \text{ mentor}, F \text{ trainee}) = a * \min[n(F \text{ trainee}), n(F \text{ mentor})] \tag{5}$$

The remaining mentors and trainees in the pool were matched randomly. Homophily was measured from the resulting population of mentor–trainee pairs. S3 Fig shows results from simulations including 1,000 mentors and 1,000 trainees.

To more clearly show change over time, the plot of temporal trends of the gender ratio of students and mentors in S6 Fig was smoothed with a 5-year moving average. For the regression analysis of temporal trends, homophily was computed from the raw data (Fig 1B and S1 Table).

For the supplemental figure of temporal trends in narrow research areas (S6 Fig), as well as comparisons of homophily across research areas, only fields with at least 1,000 students (29/73 fields with any women mentors) were included. Data from research areas with less than 1,000 students were included in analysis reported elsewhere in the paper, including aggregate statistics on homophily (see Results, S1 Table).

## Mentor seniority

Mentor seniority was measured based on the date at which the mentor began an independent academic career. When data on mentor's own graduate or postgraduate training was available,

we considered the most recent training end date as the start of the mentor's career. When no such data were available, training start date was inferred from the date of the mentor's first publication. This inference was based on a linear regression between training start date and first publication date for mentors in which both types of data were available. Because publication data were available for a much larger number of mentors than training data ($n$ = 46,368 versus 85,353), this procedure substantially increased the coverage of mentor seniority data.

## Continuation in academia

We use continuation in academia as a measurement of training outcomes. A student or postdoc has continued in academia if he or she has gone on to become a mentor (i.e., has trainees listed in the AFT database).

ProQuest does not record whether students continue to a postdoc or academic position after completing their dissertation. Given that a high percentage of PhD-level training relationships in the AFT were populated based on ProQuest (see "Data Preparation", above), this creates a risk of underestimating continuation rates. Analyses of academic continuation rates therefore only included ProQuest data for individuals whose records had been edited by at least one human contributor to the database. This criterion is based on the assumption that if individual contributors had edited a particular trainee's data, they would be likely to also record the trainees' subsequent academic position.

## Mentor status

We computed 4 metrics of a mentor's academic status:

**Trainee count.** The total number of people the mentor trained (including both PhD students and postdocs) with training end dates between 2000 and 2020.

**Funding rate.** The total funding dollars per year the mentor received in NSF and/or NIH grants. To convert total funding to rates, we divided it by the years elapsed since the first grant awarded to the mentor. Funding data were downloaded from NSF Award Search and NIH RePORTER. Grants were linked to researchers in the AFT database based on the researcher's name and institutional affiliation. Grant dollar amounts were adjusted to 2020 dollars by calculating and compensating for the linear increase in per capita funding over time. Grants awarded before 1985 were excluded due to sparse sampling of funding during this period. Mentors whose total funding exceeded $500,000,000 were excluded as outliers. A total of 20 mentors (out of 108,220) exceeded this threshold. Manual inspection suggests that mentors were in this category due to participation in large-scale, highly collaborative projects.

**H-index.** The maximum number $h$ such that the mentor has $h$ publications with at least $h$ citations and all other publications have $\leq h$ citations [29]. Citation data were drawn from the Semantic Scholar database for papers linked to researchers based on string matches to their name and the names of associated trainees and mentors [3].

**Institution rank.** The rank of the mentor's institution in the 2015–2016 Quacquarelli Symonds World University Rankings (QS rankings). Institution names in AFT were matched to university names in the QS rankings using fuzzy string matching [87]. Matches with less than 95% similarity between characters were excluded. Where the source data provided an interval rather than an exact rank, the midpoint of the interval was used as the rank for all institutions within it.

To obtain an aggregate measure of status, we calculated the first PC of trainee count, funding, $h$-index, and institution rank across mentors, which we refer to as "aggregate status." Only mentors with available data for $h$-index were included in this analysis. When other status metrics were not available, they were imputed with the mean value for that academic field. Data

for each metric were normalized by subtracting the mean and scaling to unit variance before performing principal component analysis (PCA).

To examine the relationship between mentor status and retention in academia (Figs 4, 5, S9 and S10), we fit logistic regression models predicting the probability that trainees will themselves continue to mentorship (*p*) based on their training end date (*year*), mentor seniority at the time of training end date (*seniority*), mentor and trainee gender, and mentor status:

$$log\left(\frac{p}{1-p}\right) = \beta_0 + \beta_1 \text{year} + \beta_2 \text{gender}_{\text{mentor}} + \beta_3 \text{gender}_{\text{trainee}} + \beta_4 \text{seniority} + \beta_5 \text{status}_{\text{rank}} +$$
$$\in (6)$$

where *status*$_{rank}$ indicates the mentor's rank (approximate decile) on one of the status measures discussed above relative to others in the field. To quantify the fraction of variance explained by adding mentor status (Fig 4), we fit the models above to data in which mentor status had been shuffled across trainees. The change in the marginal effect (*dy/dx*) is:

$$1 - \frac{dy/dx_{\text{actual}}}{dy/dx_{\text{shuffled}}} \qquad (7)$$

In all logistic regression models, gender was coded as "1" for men and "0" for women and the minimum training year in the dataset was subtracted from all years. Mentor seniority was converted to a 1- to 10-point scale to allow for comparison of marginal effects across models.

To examine the relationship between mentor status and homophily at the individual level, we fit linear models predicting the fraction of men students trained by each mentor based on the mentor's status and receipt of a prestigious award (see below):

$$n_{\text{men}}/n = \beta_0 + \beta_1 \text{status}_{\text{rank}} + \beta_2 \text{award} + \in \qquad (8)$$

where *status*$_{rank}$ indicates the rank (approximate decile) that the mentor's status falls into in comparison with other mentors in the field, and *award* is a categorical variable indicating whether the mentor received a Nobel, NAS membership, or HHMI funding.

We compared this to a model predicting homophily as defined in preceding analyses from mentor status:

$$\text{homophily} = \beta_0 + \beta_1 \text{status}_{\text{rank}} + \in \qquad (9)$$

Because the measurement of homophily is defined at the level of the group, we could not include award receipt as a variable in the regression model.

To identify Nobel laureates, members of the NAS, and HHMI grantees, data were drawn from official websites (http://api.nobelprize.org; http://www.nasonline.org/member-directory; https://www.hhmi.org/scientists), then linked to researcher first names using fuzzy string matching. Links with less than 95% similarity between characters were excluded. AFT profiles manually identified as Nobel laureates by contributors to the database were also included.

## Supporting information

**S1 Fig. Procedure for balancing chance of error for gender classification of men's and women's names. (A)** Dashed lines indicate upper and lower thresholds for ambiguous first names. The threshold producing balanced error is slightly more conservative for women's names (0.24) than for men's (0.75). **(B)** Distribution of inferred genders. Lists at top indicate 5 most common names in the dataset in each category (probable man, ambiguous, and probable woman). The data and code needed to generate this figure are available on Zenodo (DOI: 10.

5281/zenodo.4722020).
(TIF)

**S2 Fig. AFT dataset overview. (A)** Total graduate students, postdocs, and mentors in the AFT, during 2000–2020 and grouped by subfield. **(B)** Percentage of women graduate students in the AFT and the NSF's Survey of Earned Doctorates. Each point represents data for 2000–2020 in one broad research field. Table shows ratio difference between the 2 datasets, (NSF/AFT, absolute difference across all research areas: +0.5%). **(C)** Percentage of women postdocs in the AFT and the NSF's Statistical Survey of Graduate Students and Postdoctorates in Science and Engineering. Each point represents data for postdocs with training end dates between 2000–2020 in one broad research field (AFT data) or the 2015 cross-section of actively employed postdocs (NSF data). The data and code needed to generate this figure are available on Zenodo (DOI: 10.5281/zenodo.4722020). AFT, Academic Family Tree; NSF, National Science Foundation.
(TIF)

**S3 Fig. Homophily versus gender composition. (A)** Simulation of how measurements of gender homophily are affected by the gender composition of mentor and trainee pool within a field (see Methods). Each heatmap shows simulations for a different propensity for mentors and trainees to form same-gender pairs, ranging from 0 (none) to 1 (maximum). Scatter plot shows actual mentor and trainee gender composition for data used in the homophily analysis. Each point represents average composition for one narrow research field. When mentors or trainees of one gender are scarce, homophily does not reflect the underlying propensity to form same-gender pairs. **(B)** Simulation, after correcting for effects of gender composition of pool, with actual average compositions overlaid, as in (A). The data and code needed to generate this figure are available on Zenodo (DOI: 10.5281/zenodo.4722020).
(TIF)

**S4 Fig. Homophily in broad research areas. (A)** Bars indicate homophily in each general research area. Error bars indicate bootstrapped 95% confidence intervals. **(B)** Scatter plot compares fraction women students versus homophily in each general research area. **(C)** Fraction women mentors versus homophily, as in (B). The data and code needed to generate this figure are available on Zenodo (DOI: 10.5281/zenodo.4722020).
(TIF)

**S5 Fig. Homophily across narrow research areas. (A)** Bars indicate homophily in narrow research areas with greater than 1,000 students. Error bars indicate bootstrapped 95% confidence intervals. **(B)** Scatter plot compares fraction women students versus homophily in each narrow research area. **(C)** Fraction women mentors versus homophily, as in (B). The data and code needed to generate this figure are available on Zenodo (DOI: 10.5281/zenodo.4722020).
(TIF)

**S6 Fig. Temporal trends in gender of PhD students, split across narrow research field and mentor gender.** Colors indicate mentors' gender and time of graduation. Field abbreviations are reported in the key at lower right. The consistent pattern of homophily is reflected in the greater fraction of women trainees of women mentors (red), shifted up relative to trainees of men mentors (blue). The data and code needed to generate this figure are available on Zenodo (DOI: 10.5281/zenodo.4722020).
(TIF)

**S7 Fig. Interaction of mentor seniority and homophily.** Heatmaps show **(A)** fraction women mentors, **(B)** fraction women students, and **(C)** homophily, grouped by year that mentor

began independent career and year of student's graduation ($n = 37,692$ mentors, 163,840 students). Bars at bottom indicate results of multivariate linear regression predicting the gender-related variable from both temporal variables. Error bars indicate 95% confidence intervals. The data and code needed to generate this figure are available on Zenodo (DOI: 10.5281/zenodo.4722020).
(TIF)

**S8 Fig. Correlation (Pearson's *r*) among mentor-status metrics.** ****: $p<0.0001$, n.s.: not significant, $p \geq 0.05$. The data and code needed to generate this figure are available on Zenodo (DOI: 10.5281/zenodo.4722020).
(TIF)

**S9 Fig. Reduction in marginal effect of mentor gender after shuffling mentor status, mentor seniority, or both.** Dashed line indicates prediction of reduction for no interaction between status and seniority (i.e., the sum of the reduction from randomizing seniority and randomizing status). The data and code needed to generate this figure are available on Zenodo (DOI: 10.5281/zenodo.4722020).
(TIF)

**S10 Fig. Mentor gender effects for graduate student versus postdoctoral trainees.** Marginal effects predicted by logistic regression model on individual trainee's continuation to mentorship based on mentor status, mentor and trainee gender, and training end date, fit separately to data for postdocs and graduate students. The data and code needed to generate this figure are available on Zenodo (DOI: 10.5281/zenodo.4722020).
(TIF)

**S11 Fig. Association between gender of mentor and student and student's continuation from graduate student to mentor. (A)** Fraction of women postdocs and mentors within fields, relative to fraction of women graduate students in the field. **(B)** Proportion of graduate students in each trainee-gender/mentor-gender group across all fields, and proportion that continue to academic mentorship roles. **(C)** Fraction of graduate students within each group that continued to mentorship roles. **(D-G)** Mean difference between continuation rates for each gender group and the overall continuation rate in a given year. Error bars show jack-knifed standard error. Stars indicate significance of trainee gender (top row) or mentor gender (lower row) weights in logistic regression predicting continuation based on stop year, trainee gender, and mentor gender (****: $p<0.0001$, ***: $p<0.001$, **: $p<0.01$, *: $p<0.05$, n.s.: $p \geq 0.05$). The data and code needed to generate this figure are available on Zenodo (DOI: 10.5281/zenodo.4722020).
(TIF)

**S1 Table. Homophily across narrow research areas with at least 1,000 students.** Slope indicates annual change in homophily from 2000–2020, based on linear regression predicting homophily by year. Asterisks indicate significance of temporal effect *:$p<0.05$, **: $p<0.01$, ***: $p<0.001$. The data and code needed to generate this table are available on Zenodo (DOI: 10.5281/zenodo.4722020).
(PDF)

**S2 Table. Alternative thresholds for gender classification.** Each row shows median gender homophily across all fields and years, marginal effect of trainee and mentor gender on trainee continuation to mentorship (see Fig 3), and change in the effect of mentor gender after controlling for mentor aggregate status for a single threshold. Threshold refers to the minimum/maximum probability of a first name identifying a man for classification as a man/woman.

Excluded indicates the percent of individual excluded for having ambiguous names according to that threshold. For all thresholds, effects of mentor gender were not significant in models that accounted for status (see Fig 4). **: $p < 0.0001$, *: $p < 0.05$. Data using alternative thresholds are available from the authors on request. The code needed to generate this table is available on Zenodo (DOI: 10.5281/zenodo.4722020).
(PDF)

**S3 Table. Photo validation of automated gender estimates.** For each human-labeled category (rows), right columns indicate classification provided by genderize.io. The data and code needed to generate this table are available on Zenodo (DOI: 10.5281/zenodo.4722020).
(PDF)

**S1 Appendix. Analysis of relative continuation rate between gender groups.**
(PDF)

## Acknowledgments

We thank the members of the David Lab and Daniel E. Acuna for input.

## Author Contributions

**Conceptualization:** Leah P. Schwartz, Jean F. Liénard, Stephen V. David.

**Data curation:** Leah P. Schwartz, Jean F. Liénard.

**Formal analysis:** Leah P. Schwartz, Jean F. Liénard, Stephen V. David.

**Funding acquisition:** Stephen V. David.

**Investigation:** Leah P. Schwartz, Jean F. Liénard, Stephen V. David.

**Methodology:** Leah P. Schwartz, Stephen V. David.

**Project administration:** Stephen V. David.

**Software:** Leah P. Schwartz, Stephen V. David.

**Supervision:** Stephen V. David.

**Validation:** Leah P. Schwartz, Stephen V. David.

**Visualization:** Leah P. Schwartz, Stephen V. David.

**Writing – original draft:** Leah P. Schwartz, Stephen V. David.

**Writing – review & editing:** Leah P. Schwartz, Stephen V. David.

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
