## [Editor Report · Decision Letter 0]

26 Apr 2022

Dear Dr David, 

Thank you for submitting your revised manuscript entitled "Impact of gender on the formation and outcome of mentoring relationships in academic research" for consideration as a Research Article by PLOS Biology.

Your revisions have now been evaluated by the PLOS Biology editorial staff, and I'm writing to let you know that we would like to send your submission out for re-review.

However, before we can send your manuscript back to reviewers, we need you to complete your submission by providing the metadata that is required for full assessment. To this end, please login to Editorial Manager where you will find the paper in the 'Submissions Needing Revisions' folder on your homepage. Please click 'Revise Submission' from the Action Links and complete all additional questions in the submission questionnaire.

Once your full submission is complete, your paper will undergo a series of checks in preparation for re-review. Once your manuscript has passed the checks it will be sent out for review. To provide the metadata for your submission, please Login to Editorial Manager (https://www.editorialmanager.com/pbiology) within two working days, i.e. by Apr 28 2022 11:59PM.

Kind regards,

Roli Roberts

Roland Roberts

Senior Editor

PLOS Biology

rroberts@plos.org

---

## [Decision Letter · Decision Letter 1]

6 Jun 2022

Dear Dr David,

Thank you for your patience while we considered your revised manuscript "Impact of gender on the formation and outcome of mentoring relationships in academic research" for consideration as a Research Article at PLOS Biology. Your revised study has now been evaluated by the PLOS Biology editors, the Academic Editor, and two of the original reviewers. 

You'll see that the reviewers consider your manuscript significantly improved; however, they continue to raise a number of concerns, and in the case of reviewer #2, these are significant. When I discussed the reviews with them, the Academic Editor said, "The authors do not need to change their analysis or study design, but this will be a major rewriting of the framing and discussion / interpretation. The study has a very narrow focus and outcome so I think the suggestions of the reviewers will make it more useful."

In light of the reviews, which you will find at the end of this email, we are pleased to offer you the opportunity to address the remaining points from the reviewers in a revision that we anticipate should not take you very long. We will then assess your revised manuscript and your response to the reviewers' comments with our Academic Editor aiming to avoid further rounds of peer-review, although might need to consult with the reviewers, depending on the nature of the revisions.

**IMPORTANT - SUBMITTING YOUR REVISION**

*Resubmission Checklist*

*Published Peer Review*

*PLOS Data Policy*

*Blot and Gel Data Policy*

Sincerely,

Roli Roberts

Roland Roberts, PhD

Senior Editor

PLOS Biology

rroberts@plos.org

REVIEWERS' COMMENTS:

Reviewer #1:

In the response to my comments, the authors have provided analyses or evidence to address my concerns in previous rounds. The current manuscript is a clear improvement. Here, I have several suggestions as below:

1. The authors should take care not to generalize beyond what they are able. The title, for example, should clearly stipulate that this is focused on life sciences.

2. In addition, I am also interested to know whether the main findings generalize to other disciplines, for example social sciences and engineering. If data are limited to answer this question, the authors should discuss this limitation in the final draft.

3. A student can have multiple mentors in her/his career. For example, student A has mentors M1, M2 and M3. A-M1 is a woman-woman pair. A-M2 is a woman-man pair. And A-M3 is also a woman-man pair. A's discontinuation can be affected by all these three mentor-mentee relationships. I was wondering whether mentor fixed effect is taken into consideration. If existing data cannot support such kind of analysis, please include a limitation discussion.

Reviewer #2:

The authors' efforts to address reviewers' comments on the previous version are commendable, and the manuscript has certainly improved, including in its exploration and discussion of limitations. Nonetheless, I think there are significant issues remaining in terms of how the research and its findings are presented, starting with the abstract. I highlight some examples below, but recommend re-framing the discussion throughout the paper along these lines as well.

"Despite increasing representation in graduate training programs, a disproportionate number of women leave academic research before obtaining an independent position."

"Although trainees of women mentors are less likely to take on independent research positions..."

The outcome stated ("independent [research] position") is not what is studied in this paper. Many of us (consider your readers...) are independent academic researchers by any measure, in tenure-track faculty positions, PIs on NIH and NSF grants, mentoring many students on research including on their theses and as RAs on our sponsored research projects, but are nonetheless at institutions that do not award doctoral degrees. (Furthermore, not all PhD-granting department faculty who formally supervise doctoral theses are truly mentors, and many good mentors to a doctoral student may not be their formal PhD supervisor.) The outcome measured in this paper is whether an individual goes on to supervise (as formal advisor) a completed PhD dissertation, or possibly a postdoc. As the authors suggest, PhD supervision is an important measure as PhD advisors are training the next generation of academic faculty researchers and scientists. But the language used in the title, abstract, and throughout the paper should reflect what this study is actually examining.

Minor point: I would remove the "multidisciplinary database" references in the abstract and the text since you're focusing now on a single discipline. Similarly, in the introduction, the discussion about lack of literature comparing across fields seems out of place since the paper's looking at a single field.

Despite that, the broad field of biological sciences includes many, many subfields with different publication and citation practices, so this focus on one field doesn't fully address the possibility of gender-sorting on research topics and differences across, say, immunology versus ecology.

Counting both funding from federal agencies and number of trainees (most of whom would be supported on that funding) as status measures seems a bit like double-counting, since the number of graduate students in these fields scales as a direct function of federal research and trainee funding (see e.g. Blume-Kohout and Clack, https://journals.plos.org/plosone/article?id=10.1371/journal.pone.0082759 ). Including total trainees as a predictor of having a trainee be successful seems subject to reverse causality.

"Thus the effect of mentor gender can be explained at least partially by gender disparities in social and financial resources available to mentors."

There's no evidence of gender disparities in NIH funding outcomes conditional on applying (see the series of articles by Donna Ginther and colleagues, e.g., https://journals.plos.org/plosone/article?id=10.1371/journal.pone.0205929 ), and with federal agency funding the only measure of financial support in this study, I don't see evidence to support the conclusion that women scientists should get more financial support.

Given the option, wouldn't all PhD students prefer the advisor who is best positioned to make them successful? Should institutions assign more PhD students with institutional or fellowship funding to women faculty (regardless of the student's preference), to increase women faculty members' status? Evidence indicates biomedical sciences PhD students with traineeship or fellowship funding are less likely to take scientific research jobs at graduation, compared to students who were primarily funded as research assistants (see https://www.sciencedirect.com/science/article/pii/S004873331630035X ), so it doesn't seem like this would work. 

I'd encourage the authors to reconsider what policy implications can credibly be drawn from this research.

---

## [Editor Report · Decision Letter 2]

15 Jul 2022

Dear Dr David,

Thank you for your patience while we considered your revised manuscript "Impact of gender on the formation and outcome of formal mentoring relationships in the life sciences" for publication as a Research Article at PLOS Biology. This revised version of your manuscript has been evaluated by the PLOS Biology editors and the Academic Editor.

Based on our Academic Editor's assessment of your revision, we are likely to accept this manuscript for publication, provided you satisfactorily address the following data and other policy-related requests.

a) Please address my Data Policy requests below; specifically, we need you to supply the numerical values underlying Figs 1ABC, 2AB, 3AB, 4ABC, 5AB, S1AB, S2ABC, S3AB, S4ABC, S5ABC, S6, S7ABC, S8, S9, S10, S11ABCDEFG. I note that your Zenodo deposition currently has restricted access. Please ensure that you give us access so that we can check your compliance with the policy. If you’ve used any custom code, please also include this.

b) Please also cite the location of the data clearly in each Fig legend, e.g. “The data and code needed to generate this Figure can be found in https://zenodo.org/record/4722021”

We expect to receive your revised manuscript within two weeks. 

*Published Peer Review History*

*Press*

Sincerely,

Roli Roberts

Roland Roberts, PhD

Senior Editor,

rroberts@plos.org,

PLOS Biology

DATA POLICY:

Regardless of the method selected, please ensure that you provide the individual numerical values that underlie the summary data displayed in the following figure panels as they are essential for readers to assess your analysis and to reproduce it: Figs 1ABC, 2AB, 3AB, 4ABC, 5AB, S1AB, S2ABC, S3AB, S4ABC, S5ABC, S6, S7ABC, S8, S9, S10, S11ABCDEFG. NOTE: the numerical data provided should include all replicates AND the way in which the plotted mean and errors were derived (it should not present only the mean/average values).

DATA NOT SHOWN?

---

## [Editor Report · Decision Letter 3]

27 Jul 2022

Dear Dr David,

Thank you for the submission of your revised Research Article "Impact of gender on the formation and outcome of formal mentoring relationships in the life sciences" for publication in PLOS Biology. On behalf of my colleagues and the Academic Editor, Lisa Bero, I am happy to say that we can in principle accept your manuscript for publication. Please address any remaining formatting and reporting issues that will be detailed in an email you should receive within 2-3 business days from our colleagues in the journal operations team; no action is required from you until then. Please note that we will not be able to formally accept your manuscript and schedule it for publication until you have completed any requested changes.

PRESS

Sincerely, 

Roli Roberts

Senior Editor

PLOS Biology

rroberts@plos.org